# Short-Term Effects of Brolucizumab in the Treatment of Wet Age-Related Macular Degeneration or Polypoidal Choroidopathy Refractory to Previous Anti-Vascular Endothelial Growth Factor Therapy

**DOI:** 10.3390/medicina58121703

**Published:** 2022-11-22

**Authors:** Alan Y. Hsu, Chih-Ying Lin, Chun-Ju Lin, Chun-Ting Lai, Ning-Yi Hsia, Jane-Ming Lin, Peng-Tai Tien, Ping-Ping Meng, Wei-Ning Ku, Wen-Lu Chen, Yi-Yu Tsai

**Affiliations:** 1Department of Ophthalmology, China Medical University Hospital, China Medical University, 2 Yuh-Der Road, Taichung 40447, Taiwan; 2Department of General Medicine, China Medical University Hospital, Taichung 406040, Taiwan; 3School of Medicine, College of Medicine, China Medical University, Taichung 406040, Taiwan; 4Department of Optometry, Asia University, Taichung 413305, Taiwan; 5Graduate Institute of Clinical Medical Science, College of Medicine, China Medical University, Taichung 406040, Taiwan

**Keywords:** anti-vascular endothelial growth factor therapy, brolucizumab, pigment epithelial detachment, polypoidal choroidopathy, subretinal fluid, wet age-related macular degeneration

## Abstract

*Background and objectives*: To report the initial response to a single intravitreal brolucizumab (IVI-B) injection in wet age-related macular degeneration (wAMD) or polypoidal choroidopathy (PCV) complicated with either persistent subretinal fluid (SRF) or pigment epithelial detachment refractory to previous anti-vascular endothelial growth factor (anti-VEGF) therapy. *Material and methods*: In this retrospective study, all eyes received a single IVI-B (6 mg/0.05 mL) for wAMD or PCV with treatment-resistant SRF or PED. Outcome measures included assessment in central retinal thickness (CRT), visual acuity, and evaluation for changes in the SRF or PED on OCT. Follow-up was prior to the first brolucizumab injection, then at 1 week and 5 weeks afterwards. *Results*: In total, 10 eyes of 10 patients (6 women [60%]) were enrolled. Five patients had wAMD and five patients had PCV. Average age of participants was 67.6 years. All patients received one IVI-B. All patients were not treatment-naïve to anti-VEGF agents. At the first week and fifth week following the first IVI-B, seven out of seven patients (100%) had resolved SRF. However, seven out of nine patients (78%) had no improvement of their PED at 5 weeks follow-up. Mean PED height and width before the first IVI-B was 339.77 µm and 2233.44 µm, respectively. Mean PED height and width at the fifthweek following the first IVI-B was 328.125 µm and 2129.5 µm, respectively. Overall mean visual acuity before the first IVI-B was 0.224; and 5 weeks following the first IVI-B was 0.38. *Conclusions*: Treatment with brolucizumab resulted in anatomical improvement for all patients with persistent SRF. Limited efficacy was seen for persistent PED. Brolucizumab appears to be a safe and effective option for treatment-resistant SRF. Future multicenter collaborative studies are warranted.

## 1. Introduction

Intravitreal (IVI) anti-vascular endothelial growth factor (anti-VEGF) injections are used to treat retinal diseases [1] including wet age-related macular degeneration (wAMD), choroidal neovascularization, polypoidal choroidal vasculopathy (PCV) as well as macular edema secondary to other causes [2]. One recent example of a new anti-VEGF agent is Brolucizumab (Novartis, East Hanover, NJ, USA) (IVI-B), an FDA-approved treatment for wAMD [3]. Most literature so far has explored the long-term effects of this anti-VEGF agent and after multiple injections [4,5]. However, currently there is a paucity of research that has looked into the short-term effects of Brolucizumab and after a single injection on treatment-resistant wAMD or PCV patients. We report the short-term response of this new anti-VEGF treatment on either persistent subretinal fluid (SRF) or pigment epithelial detachment [6] refractory to previous anti-VEGF therapy among patients with either wAMD or PCV.

## 2. Materials and Methods

### 2.1. Methods

This is a single center, retrospective study performed at the China Medical University Hospital (CMUH), Taiwan. A total of 10 patients with either wAMD or PCV with persistent SRF or pigment PED who underwent previous treatment with anti-VEGF agents were reviewed. Best-corrected visual acuity (BCVA) assessment, central retinal thickness (CRT), and functionally relevant anatomic findings, such as SRF and PED, were retrospectively retrieved from electronic records. CRT were evaluated via spectral domain OCT (SD-OCT) device (Heidelberg Engineering, Heidelberg, Germany) and defined as the mean thickness of the 1000 μm central diameter area. All OCT findings including maximum height and width of the PED were evaluated using Heidelberg Eye Explorer Software (v. 5.6.4.0; Heidelberg Engineering, Heidelberg, Germany). All patients were followed up at the 1st week and 5th week after single injection of IVI-B.

Institutional Review Board have approved the study protocol (CMUH110-REC3-101) (10 August 2021) and informed consent was obtained from all participants. The study complies with the tenets of the Declaration of Helsinki.

### 2.2. Inclusion Criteria

We included eyes with diagnosed wAMD or PCV treated with only one injection of brolucizumab who had follow-ups at 1 week and 5 weeks with spectral domain OCT (SD-OCT) device (Heidelberg Engineering, Heidelberg, Germany).

### 2.3. Exclusion Criteria

Exclusion criteria were the presence of macular disease other than wAMD or PCV or any history of vitrectomy.

## 3. Results

### 3.1. Characteristics of the Study Population

In total, 10 eyes of 10 patients met the inclusion criteria and were recruited. Baseline characteristics of the study population are summarized in Table 1 and Table 2. Overall, 16 patients (60%) were female with the mean age being 67.6 years old. In terms of co-morbidities, eight patients (80%) had hypertension, four patients (40%) had diabetes and one patient (10%) was a smoker. In terms of previous intra-ocular therapies received, 10 eyes (100%) had been treated with other available anti-VEGF agents. The mean number of anti-VEGF injections received prior to the IVI-B for all 10 eyes was 14.3. Two eyes (20%) had received previous photodynamic therapies (PDT). Four patients (40%) had phakic status. In terms of presenting complications, seven eyes (70%) had SRF prior to the first IVI-B, while nine eyes (90%) had PED prior to first IVI-B.

### 3.2. Anatomical and Functional Outcomes

The changes in OCT findings for all 10 eyes at baseline before intra-vitreal injection of brolucizumab (IVI-B), 1 week after IVI-B and 5 weeks after IVI-B, are illustrated in Appendix A. The anatomical and functional outcomes for all patients (*n* = 10), wAMD patients (*n* = 5) and PCV patients (*n* = 5) are summarized in Table 2, Table 3 and Table 4, respectively.

In terms of the overall changes in mean visual acuity (VA), the overall mean visual acuity at baseline before IVI-B was 0.224; and the mean VA at 5 weeks after injection was 0.38 (Table 2). Table 3 showed the mean VA for the wAMD cohort at baseline before IVI-B was 0.228; and 0.4 at 5 weeks after injection. Table 4 showed the mean VA at baseline was 0.22; and 0.36 at 5 weeks after injection.

In terms of the overall changes in central retinal thickness (CRT), the overall baseline mean CRT was 367.9 μm; 317 μm at 1 week after IVI-B; and 265.2 μm at 5 weeks after IVI-B, as seen in Table 2. Table 3 showed the mean CRT for the wAMD cohort was 283.8 μm at baseline before IVI-B; 241.4 μm at 1 week after IVI-B; and 248.5 μm at 5 weeks after IVI-B. Table 4 showed that the mean CRT for the PCV cohort was 452 μm at baseline before IVI-B; 392.6 μm at 1 week after IVI-B and 331.8 μm at 5 weeks after IVI-B

In terms of pigment epithelial detachment (PED) status (Table 2, Table 3 and Table 4), nine eyes had PED at baseline before IVI-B. Out of the initial nine eyes with PED, four eyes and five eyes had concurrent wAMD and PCV, respectively. At 5 weeks after IVI-B, two eyes (22%) overall had improved PED compared to baseline. Out of the two eyes that had improved PED compared to baseline, 100% of them had concurrent PCV; 0% of the wAMD with PED at baseline before IVI-B had improved PED at 5 weeks after injection.

Concerning the mean PED height and width, the overall mean PED height (see Table 2) at baseline before IVI-B was 339.77 μm; 311.11 μm at 1 week after IVI-B; and 328.125 μm at 5 weeks after IVI-B. The overall mean PED width (see Table 2) at baseline before IVI-B was 2233.44 μm; 2168.77 μm at 1 week after IVI-B; and 2129.5 μm at 5 weeks after IVI-B. The mean PED height for the wAMD cohort (see Table 3) at baseline before IVI-B was 293 μm; 273 μm at 1 week after IVI-B; and 271 μm at 5 weeks after IVI-B. The mean PED width for the wAMD cohort (see Table 2) at baseline before IVI-B was 2165.5 μm; 2087 μm at 1 week after IVI-B; and 1425.3 μm at 5 weeks after IVI-B. The mean PED height for the PCV cohort (see Table 4) at baseline before IVI-B was 377.2 μm; 757.8 μm at 1 week after IVI-B; and 749.4 μm at 5 weeks after IVI-B. The mean PED width for the PCV cohort (see Table 2) at baseline before IVI-B was 2383.81 μm; 1818 μm at 1 week after IVI-B; and 2165 μm at 5 weeks after IVI-B.

In terms of subretinal fluid (SRF) status (Table 2, Table 3 and Table 4), seven eyes overall had SRF at baseline before IVI-B. Out of all the seven eyes that had SRF initially, five eyes and two eyes had concurrent wAMD and PCV, respectively. At 5 weeks follow-up after IVI-B, seven eyes (100%) had improved SRF compared to baseline. Out of the seven eyes that had improved SRF compared to baseline, five eyes had concurrent wAMD and two eyes had concurrent PCV.

## 4. Discussion

To our knowledge, this is one of the few reports so far that have evaluated the short-term real-life clinical response of IVI-B for the treatment on either persistent SRF or PED refractory to previous anti-VEGF therapy among patients with either wAMD or PCV.

### 4.1. Novel Findings

In the present study, anatomical improvements towards SRF can be seen after a single injection of brolucizumab. Limited effects, however, were seen after a single IVI-B towards PED in our study. We found that at follow-up at 1 week and 5 weeks after the first IVI-B, seven out of seven patients (100%) had resolution of SRF. Out of the seven eyes with improved SRF; five of the eyes had wAMD and two of the eyes had PCV. In terms of PED, only two patients (22%) had reduced PED. Both of these patients had concurrent PCV. Unfortunately, no improvement in the PED was seen in seven patients (78%) at 5 weeks following the first IVI-B. Mean PED height and width at baseline was 339.77 μm and 2233.44 μm, respectively. At 5 weeks duration after the first IVI-B, mean PED height and width decreased to 328.125 μm and 2129.5 μm, respectively (see Table 3). Additionally, we also demonstrated a reduction in mean CRT from 367.9 μm at baseline before injection to 265.2 μm at 5 weeks after injection. Lastly, limited functional effects from IVI-B were noted in our study, with only minor improvements in overall mean VA seen at 5 weeks follow-up after IVI-B compared to baseline.

### 4.2. Clinical Implications

Brolucizumab is an antibody fragment that is able to bind to all human isoforms of anti-vascular endothelial growth factor (VEGF) A [7]. To date, short-term outcomes and single intravitreal injection studies are limited. A few notable studies on IVI-B include these three landmark studies: HAWK, HARRIER and the Merlin studies; all of which demonstrated non-inferiority of brolucizumab compared to a conventional anti-VEGF agent (aflibercept) after multiple intravitreal injections and its effect on anatomical outcomes after a few months of follow-up [4,5]. We further contribute by providing real-world data on its short-term effects and after a single IVI-B. In addition, our results are unique because the study is based on treatment-resistant eyes that are refractory to previous anti-VEGF injections. Major studies on brolucizumab so far like HAWK and HARRIER [4] were based on untreated eyes that have not received any anti-VEGF injections before. Better understanding of how IVI-B works on treatment refractory patients is important, as PCV and wAMD are considered chronic disorders that require long-term follow-up and treatment. It has been reported in a 7-year follow-up study on AMD patients who received anti-VEGF injections, that 50% of the patients still required some form of treatment at the seventh year of follow-up [8]. Treatment-refractory PCV and wAMD are therefore a clinically challenging subset of patients that clinicians may encounter often in practice. The implications from our results are that it would allow for a better understanding of the short-term profiles of a single injection of IVI-B for this treatment-resistant clinical cohorts of patients through real-world data.

### 4.3. Comparison to Other Studies

One major finding was that we have shown great short-term response towards SRF by single IVI-B. This is in comparison to a study by Awh et al., where a reduction in most of their SRF was seen after 6 months of follow-up [9]. Our result complements this by showing a reduction in SRF in 100% of our cases and with only one injection, and at a shorter duration of follow-up.

Another outcome was the response on PED. We found that only two out of nine patients with PED were found responsive to IVI-B at the 5th week of follow-up. However, reduction in the overall mean PED height and width at 5 weeks following the first IVI-B was also seen. Though some studies have shown a significant reduction in width and height of PED after IVI-B treatment [10,11], other studies have also shown resolution of PED after multiple IVI-B therapies [12]. We postulate the potential reasons behind the poor response of our PED towards IVI-B as follows. Firstly, we are unable to exclude the possibility that our case series is predominantly made up of a more unresponsive type of PED. PED of fibrovascular subtype tended to respond poorly to anti-VEGF treatment [13,14]. The reason is unknown but possibly involves its actual physical make-up in comparison to other subtypes [15]. With brolucizumab being an anti-VEGF-based agent, it seems plausible that this poor response extends to this new agent. This is implied in studies where greater therapeutic response towards brolucizumab is seen when more favorable subtypes of PED were recruited (such as the serous subtype) [10]. On a further note, the two cases of PED that improved at 5 weeks after IVI-B were those from the PCV cohort. Though wAMD and PCV are conditions that share common environmental and genetic determinants, they each have their own natural history and unique responses towards certain therapies [16]. Therefore, it is possible that the concurrent conditions of wAMD or PCV could have played a role on the resultant PED in question. However, on the other hand, the seven cases that had PED which did not improve involved four wAMD and three PCV cases. From this, it can be seen that there is an almost even distribution of the two different cohorts (wAMD vs. PCV) among the PED that did not improve; so, the role the concurrent conditions of PCV or wAMD played in the overall therapeutic response of the PED remains elusive.

Secondly, baseline characteristics might also be another explanation. Multiple studies have confirmed that elderly age, previous intraocular therapies such as receiving previous anti-VEGF injections, having medical comorbidities, as well as smoker status may all play a role in the therapeutic outcomes of those undergoing anti-VEGF therapies [17,18,19]. In terms of age, the mean age in our study was 67.6 years old. Elderly patients tend to have various medical illnesses, and this is reflected in the comorbidities recorded in our study, where 80% and 40% of our study participants had hypertension and diabetes, respectively. These combinations of factors including smoking have been shown to promote retinal hypoxia. This leads to increased serum VEGF levels [19] and higher VEGF vitreous concentrations have also been associated with worse outcome. All these could contribute to the wAMD and PCV recruited in our study to possibly be more resistant towards any anti-VEGF injections such as IVI-B. Additionally, history of previous intra-ocular therapies such as previous anti-VEGF injections may have also been a factor. Studies have shown that multiple anti-VEGF agents over a period of time can reduce normal mediators in the eye necessary for the maintenance of normal retinal structures. This can contribute to a more treatment-resistant wAMD and PCV [20]. Future studies into IVI-B could recruit a suitable control group with shared demographics as well as not possess any of the confounders mentioned.

Thirdly, another point to be highlighted is ethnicity. It has been increasingly recognized that distinct differences in clinical phenotype of PCV and wAMD can be seen between Caucasian and non-Caucasian eyes. For example, Asians with PCV and AMD tend to present earlier and tend to have larger lesions involving the fovea compared to Caucasians [21,22]. However, whether this translates into actual differences in responses towards anti-VEGF therapies such as IVI-B is unknown. Interestingly, in a study of 1089 AMD patients by Javaheri et al. which consisted mostly of Caucasians, 35.5% of their overall patients demonstrated a significant improvement in PED at the first month after anti-VEGF injection. They also showed that out of all their AMD patients who had improved PED; only 1 was Asian (0.5% of their overall study participants), while the other 97.1% of the responsive patients were white [23]. Our findings in comparison—with only two out of seven of our PED showing any improvements after IVI-B, were based on an ethnic make-up of entirely an Asian Taiwanese study population. Therefore, our data along with other published studies seem to raise the possibility that the poor response in our PED could be influenced by ethnicity as well.

Fourthly, it is perhaps due to our short-term follow-up; longer follow-up or additional injections of brolucizumab might show a more favorable outcome.

### 4.4. Strengths and Limitations

Our strength of this study includes assessing for the real-world, short-term clinical response of a single IVI-B among patients who are treatment refractory towards other anti-VEGF injections and this makes our results more generalizable compared to studies with clinical trial designs.

Unfortunately, our study was limited by our small injection number, small sample size, limited follow-up, lack of treatment naïve eyes, and lack of ethnic diversity. These would have implications about the generalizability of our results. It should be highlighted that our short-term follow-up of up to 5 weeks after IVI-B, though classified as a limitation, was one of our study aims and can also be seen as a strength. Furthermore, our study was based at a tertiary referral medical center for ophthalmology. Therefore, referral bias should also be considered.

## 5. Conclusions

From this retrospective study, anatomical improvement was seen in terms of subretinal fluid in all of our patients during 5 weeks follow-up after a single injection of brolucizumab. Limited therapeutic benefit towards PED was also seen. Although our results from IVI-B may initially appear unimpressive, this was an expected outcome coming from study participants that were treatment refractory to alternative anti-VEGF agents. In summary, we demonstrated the short-term clinical effectiveness of brolucizumab IVI on wAMD or PCV patients with treatment-resistant SRF.

## Figures and Tables

**Table 1 medicina-58-01703-t001:** Baseline characteristics of all patients (*n* = 10).

Variables	wAMD Eyes, *n* = 5	PCV Eyes, *n* = 5	All Eyes, *n* = 10
Eye (right eye, %)	2, 40%	4, 80%	6, 60%
Age, Mean (years)	68.4	66.8	67.6
Hypertension (eyes, %)	5, 100%	3, 60%	8, 80%
Diabetes (eyes, %)	3, 60%	1, 20%	4, 40%
Smoker (eyes, %)	1, 20%	0, 0%	1, 10%
Mean number of previous Anti-VEGF injections (no.)	15.2	13.4	14.3
Phakic (%)	40%	40%	40%
Previous PDT (%)	0%	40%	20%
Baseline ^a^ SRF (eyes, %)	5, 100%	2, 40%	7, 70%
Baseline PED (eyes, %)	4, 80%	5, 100%	9, 90%

PED, pigment epithelial detachment; PCV, polypoidal choroidal vasculopathy; PDT, Photodynamic Therapy; SRF, subretinal fluid; VEGF, vascular-endothelial growth factor; wAMD, wet age-related macular degeneration. ^a^ Baseline defined as the time immediantly prior to the first injection of brolucizumab.

**Table 2 medicina-58-01703-t002:** Anatomical and visual outcomes in all patients at baseline before injection, 1 week and 5 weeks post-injection (all patients, *n* = 10).

Variables	Baseline Prior to Brolucizumab Injection	1 Week after Brolucizumab Injection	5 Weeks after Brolucizumab Injection
Mean VA	0.224	-	0.38
Mean CRT (μm)	367.9	317	265.2
SRF at baseline (eyes)	7	-	-
PED at baseline (eyes)	9	-	-
Improved ^b^ SRF at 5 weeks follow-up post-injection (eyes, %)	-	-	7, 100%
Improved PED at 5 weeks follow-up post-injection (eyes, %)	-	-	2, 22%
Mean PED height (μm)	339.77	311.11	328.125
Mean PED width (μm)	2289	2168.77	2129.5

-, not applicable to this case; CRT, central retinal thickness; PED, pigment epithelial detachment; SRF, subretinal fluid. ^b^ Improvement defined as the reduction or resolution of the indicated complication seen on OCT.

**Table 3 medicina-58-01703-t003:** Anatomical and visual outcomes in wAMD patients at baseline before injection, 1 week and 5 weeks post-injection (wAMD only: *n* = 5).

Variables	Baseline Prior to Brolucizumab Injection	1 Week after Brolucizumab Injection	5 Weeks after Brolucizumab Injection
Mean VA	0.228	-	0.4
Mean CRT (μm)	283.8	241.4	248.5
SRF at baseline (eyes)	5	-	-
PED at baseline (eyes)	4	-	-
Improved SRF at 5 weeks follow-up post-injection (eyes, %)	-	-	5, 100%
Improved PED at 5 weeks follow-up post-injection (eyes, %)	-	-	0, 0%
Mean PED height (μm)	293	273	271
Mean PED width (μm)	2165.5	2087	1425.3

**Table 4 medicina-58-01703-t004:** Anatomical and visual outcomes in PCV patients at baseline before injection, 1 week and 5 weeks post-injection (PCV only: *n* = 5).

Variables	Baseline Prior to Brolucizumab Injection	1 Week after Brolucizumab Injection	5 Weeks after Brolucizumab Injection
Mean VA	0.22	-	0.36
CRT (μm)	452	392.6	331.8
SRF at baseline (eyes)	2	-	-
PED at baseline (eyes)	5	-	-
Improved SRF at 5 weeks post-injection (eyes, %)	-	-	2, 100%
Improved PED at 5 weeks post-injection (eyes, %)	-	-	2, 40%
Mean PED height (μm)	377.2	341.4	362.4
Mean PED width (μm)	2383.81	2234	2552

## Data Availability

The dataset used for analysis is available from the corresponding authors Chun-Ju Lin under reasonable request. All data used were included as tables and graphs in current study.

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
