# Peer review of "Short-Term Effects of Brolucizumab in the Treatment of Wet Age-Related Macular Degeneration or Polypoidal Choroidopathy Refractory to Previous Anti-Vascular Endothelial Growth Factor Therapy"

_medicina, 2022, doi:10.3390/medicina58121703_

Round 1

Reviewer 1 Report

Dear authors, your manuscript presents an interesting and potentially important clinical result showing some significant anatomical changes after treatment with brolucizumab. The pigment epithelium detachment area sustained a big decrease after the treatment (anatomical changes). It remains to be seen if this translates into functional changes (vision improvement). The absence of these data reduces the overall impact of this report.  However, knowing that the extent of RPE detachment positively correlates with loss of vision during wet AMD, it is expected that the above treatment has a chance to achieve a positive result while treating wet AMD. The study is well written and has merit to be published. You must follow up with a report of a longer-term study to describe any functional impact on vision

Author Response

Dear Reviewer,

Thank you for giving us the opportunity to submit a revised draft of my manuscript. We have carefully addressed points raised and have been able to incorporate changes to reflect the suggestions provided by the reviewers.

Comments from Reviewer 1

Comment 1: Dear authors, your manuscript presents an interesting and potentially important clinical result showing some significant anatomical changes after treatment with brolucizumab. The pigment epithelium detachment area sustained a big decrease after the treatment (anatomical changes). It remains to be seen if this translates into functional changes (vision improvement). The absence of these data reduces the overall impact of this report.  However, knowing that the extent of RPE detachment positively correlates with loss of vision during wet AMD, it is expected that the above treatment has a chance to achieve a positive result while treating wet AMD. The study is well written and has merit to be published. You must follow up with a report of a longer-term study to describe any functional impact on vision

Thank you for pointing this out and we agree with your points raised. We certainly hope we will be able to conduct such a study in the future.

With regards to the functional impact on vision, we have modified the results section slightly (see line 87-90) to demonstrate the changes in mean visual acuity at baseline and at 5 weeks after brolucizumab injection for all patients as well as for the wAMD and PCV cohort. We have also modified the tables accordingly for better comparison between clinical groups (wAMD vs PCV) as well as a table for overall changes in anatomical and visual outcomes.

In terms of the follow up period, our study aim was to look at the short-term outcomes of IVI-B among treatment-resistant patients and our chosen follow up period hopefully reflects this. Our study aim was chosen because there is a lack of studies currently that looked at this, as emphasized in line 181-183. 

Reviewer 2 Report

The authors report Short Term Effects of Brolucizumab in wAMD or PCV refractory to previous anti-vascular endothelial growth factor (anti-VEGF) therapy.

1.      Table 1 should be adjusted to a more readable format, and displayed on one page.

2.      Result section was more like a serial cases report. Since the study include two different diseases, it will be better to classify, compare and summarize the results.

3.      Patients information such as systemic diseases, smoking status, medical history and intro-ocular treatment history etc. was not mentioned. Were these factors related to the treatment outcome?

Author Response

Dear Reviewer,

Thank you for giving us the opportunity to submit a revised draft of my manuscript. We have carefully addressed points raised and have been able to incorporate changes to reflect the suggestions provided by the reviewers.

Comments from Reviewer 2

 Comment 1:  The authors report Short Term Effects of Brolucizumab in wAMD or PCV refractory to previous anti-vascular endothelial growth factor (anti-VEGF) therapy.

Table 1 should be adjusted to a more readable format, and displayed on one page. Result section was more like a serial cases report. Since the study include two different diseases, it will be better to classify, compare and summarize the results. Patients information such as systemic diseases, smoking status, medical history and intro-ocular treatment history etc. was not mentioned. Were these factors related to the treatment outcome?

 Thank you for these suggestions and we agree completely. Specific modifications based on the reviewer’s suggestions have been made which include changes made to our tables. In terms of tables, we have simplified and broken down the original table as suggested into 4 different tables. Table 1 is the overall characteristics of the patients, of which includes systemic diseases, smoking status and intra-ocular treatments (like previous anti-VEGF injections, phakic status and photodynamic therapies). Table 2 are to outline the outcomes from the follow ups of all our patients. Table 3 and Table 4 are for outcomes from the follow ups of the wAMD and the PCV cohort, respectively. This would allow for better comparison between clinical groups, as per the reviewer’s suggestion.

We have also modified the results sections as well, breaking down the results into outlining the overall outcome changes in follow ups as well as the individuals changes from the wAMD and the PCV cohort (line 81-116) so that our manuscript woud read less like a serial case report. We have also moved the original two figures demonstrating the OCT findings of all ten of our cases into the supplementary material (Figure S1 and Figure S2) to not overcomplicate the result section while also providing a point of reference for reviewers if they so wish to evaluate the corresponding OCT images.

In terms of whether these baseline factors influenced our treatment outcome, we outlined our analysis from line 208 to 219. In brief, we believe some of these factors might have played a role in the poor response of PED shown towards brolucizumab. We have also reviewed the literature to support our analysis.

We also took initiative and looked at what role the baseline ethnicity might have played (if at all) in terms of the therapeutic response seen in our study (line 221-230).

Reviewer 3 Report

The authors described the short term effects of brolucizumab for the eyes with refractory to the other anti-VEGF drug.

However this study include very small sample size (10 eyes) and the follow-up period is as short as 5 weeks. It has been widely known that AMD is a disease for which long-term management is important, and we believe that results from a single dose alone are not very meaningful. The author thinks that the results should be reviewed again with a larger number of patients and a longer observation period.

Author Response

Dear Reviewer 3,

Thank you for giving us the opportunity to submit a revised draft of my manuscript. We have carefully addressed points raised and have been able to incorporate changes to reflect the suggestions provided by the reviewers.

Comments from Reviewer 3

Comment 1: The authors described the short term effects of brolucizumab for the eyes with refractory to the other anti-VEGF drug. However this study include very small sample size (10 eyes) and the follow-up period is as short as 5 weeks. It has been widely known that AMD is a disease for which long-term management is important, and we believe that results from a single dose alone are not very meaningful. The author thinks that the results should be reviewed again with a larger number of patients and a longer observation period.

Thank you for your suggestion and we do agree with the certain points mentioned. As seen in line 250-252, despite our unimpressive results towards brolucizumab - which were expected from such a treatment-refractory cohort of patients, the strength of our study lies in the fact that we aimed to fill in the gaps in the current literature on Brolucizumab. Currently, several studies have already looked at multiple injections of brolucizumab on treatment-naïve eyes and their outcome through long-term follow ups. Despite our limitations (including the ones highlighted by reviewer 3), our study still seek to contribute by being one of the first studies in providing real-world results on the short-term outcomes of a single injection of brolucizumab among treatment-resistant patients. These thoughts are emphasized in line 181-183. 

Round 2

Reviewer 2 Report

In their revised manuscript, the authors have adequately responded to the comments provided in the original reviews. They also included more discussions for helping comprehensively explain the results. But there are still some issues need to be addressed. 

1.      All tables should be showed as three-line tables.

2.  It will be better to include individual patient information in Supplementary data, corresponding with the Supplementary figures.

Author Response

Dear Reviewers,

Thank you for giving us the opportunity to further submit a revised draft of our manuscript based on your valuable suggestions.

Comments from Reviewer 2

Comment 1:   All tables should be showed as three-line tables.

Thank you for this suggestion and we agree completely. We have further modified the tables per your suggestions to three-line tables for easier reading.

Comment 2:   It will be better to include individual patient information in Supplementary data, corresponding with the Supplementary figures.

Thank you for this suggestion and we agree completely. Per the reviewer’s suggestion, in the supplementary materials, we have added the corresponding table that includes all the individual patients’ information (baseline, 1st week and 5th week follow up data) below the supplementary figures. We have also modified the table to be a three-line table as well, in keeping with the reviewer’s excellent suggestion. We have also taken the initiative to make  small modifications to the captions of supplementary Figure S1 and S2 to clarify which OCT images correspond to which patient.

Reviewer 3 Report

The authors have revised this manuscript well and it is a good text, but considering the limitations of the small number of cases (10 eyes) and the variable observation period after a single dose, it is not considered to be worthy.

Author Response

Dear Reviewer,

Thank you for giving us the opportunity to submit a revised draft of my manuscript based on your valuable suggestions.

Comments from Reviewer 3

Comment 1: The authors have revised this manuscript well and it is a good text, but considering the limitations of the small number of cases (10 eyes) and the variable observation period after a single dose, it is not considered to be worthy.

Thank you for your input and we appreciate the time the reviewer took in evaluating our manuscript. We do agree with the reviewer in terms of the small case numbers and have acknowledged it in our limitations (line 273). However, we do sincerely hope the reviewers will take into account our other merits as well. This includes our study’s primary strength – which is, being one of the first to demonstrate short-term clinical effectiveness of a single injection of brolucizumab IVI on treatment-resistant SRF (line 269-272).
